# Regulatory Roles of Flavonoids in Caspase-11 Non-Canonical Inflammasome-Mediated Inflammatory Responses and Diseases

**DOI:** 10.3390/ijms241210402

**Published:** 2023-06-20

**Authors:** Young-Su Yi

**Affiliations:** Department of Life Sciences, Kyonggi University, Suwon 16227, Republic of Korea; ysyi@kgu.ac.kr; Tel.: +82-31-249-9644

**Keywords:** flavonoid, non-canonical inflammasome, anti-inflammatory, inflammatory disease, nutraceutical

## Abstract

Inflammasomes are multiprotein complexes that activate inflammatory responses by inducing pyroptosis and secretion of pro-inflammatory cytokines. Along with many previous studies on inflammatory responses and diseases induced by canonical inflammasomes, an increasing number of studies have demonstrated that non-canonical inflammasomes, such as mouse caspase-11 and human caspase-4 inflammasomes, are emerging key players in inflammatory responses and various diseases. Flavonoids are natural bioactive compounds found in plants, fruits, vegetables, and teas and have pharmacological properties in a wide range of human diseases. Many studies have successfully demonstrated that flavonoids play an anti-inflammatory role and ameliorate many inflammatory diseases by inhibiting canonical inflammasomes. Others have demonstrated the anti-inflammatory roles of flavonoids in inflammatory responses and various diseases, with a new mechanism by which flavonoids inhibit non-canonical inflammasomes. This review discusses recent studies that have investigated the anti-inflammatory roles and pharmacological properties of flavonoids in inflammatory responses and diseases induced by non-canonical inflammasomes and further provides insight into developing flavonoid-based therapeutics as potential nutraceuticals against human inflammatory diseases.

## 1. Introduction

Inflammasomes are multiprotein complexes comprised of a pattern recognition receptor (PRR) and inflammatory molecules that provide the molecular platforms of inflammatory responses in response to various pattern-associated molecular patterns (PAMPs) and danger-associated molecular patterns (DAMPs) [1,2]. There are two main classes of inflammasomes: canonical and non-canonical. Canonical inflammasomes include nucleotide oligomerization domain-like receptor (NLR) family inflammasomes, such as NLRP1, NLRP3, NLRC4, NLRP6, NLRP9, and NLRP12, and non-NLR family inflammasomes, such as pyrin and absent in melanoma 2 (AIM2) inflammasomes [1,2]. Numerous studies have successfully demonstrated that canonical inflammasomes activate inflammatory responses under the stimulation of PAMPs and DAMPs, leading to the onset and progression of various inflammatory diseases [3,4,5,6,7,8]. Other types of inflammasomes have recently been identified, including human caspase-4, caspase-5, and mouse caspase-11 inflammasomes, which were named non-canonical inflammasomes because they have similar roles to but are distinguished from canonical inflammasomes [9,10,11,12]. Lipopolysaccharide (LPS) has been identified as the only PAMP that activates non-canonical inflammasomes via direct interaction with caspase-4/5/11 [13,14,15,16]. Activation of non-canonical inflammasomes induces the proteolytic cleavage of gasdermin D (GSDMD), and the amino-terminal fragments of GSDMD (N-GSDMD) generate GSDMD pores in cell membranes, resulting in pyroptosis, an inflammatory form of cell death [13,14,15,16]. Activation of non-canonical inflammasomes also induces proteolytic activation of caspase-1, but unlike canonical inflammasomes, they indirectly activate caspase-1. They directly activate the NLRP3 canonical inflammasome, and the activated NLRP3 canonical inflammasome induces the proteolytic activation of caspase-1, which suggests that canonical and non-canonical inflammasomes play a cooperative role in activating inflammatory signaling pathways [17,18,19,20]. Activated caspase-1 subsequently promotes proteolytic maturation and secretion of the pro-inflammatory cytokines interleukin (IL)-1β and -18 through GSDMD pores [13,14,15,16]. Emerging studies have demonstrated that non-canonical inflammasomes also play critical roles in inflammatory responses and in numerous infectious and inflammatory diseases [21,22,23,24,25,26,27,28,29,30,31,32,33,34,35,36,37,38].

Flavonoids are a group of natural ingredients, particularly secondary metabolites, widely found in fruits, vegetables, grains, nuts, seeds, bark, roots, stems, flowers, teas, and wine, and are essential for humans to improve health, increase longevity, and promote immunity [39]. Flavonoids play numerous pharmacological roles in various human diseases, including cancers and infectious, cardiovascular, neurodegenerative, respiratory, allergic, and metabolic diseases [40,41,42,43,44,45,46]. Considerable efforts have also been made to demonstrate the anti-inflammatory role of flavonoids in inflammatory responses and diseases [47,48,49]; however, most studies have focused on priming, the preparation step of inflammatory responses [50,51,52,53,54]. Recent studies have also reported the anti-inflammatory role of flavonoids by targeting inflammasome activation in the triggering step of inflammatory responses [49,55,56,57,58,59]. Interestingly, growing evidence has demonstrated that flavonoids also have anti-inflammatory actions and alleviate various inflammatory diseases by inhibiting non-canonical inflammasomes in the triggering step of inflammatory responses. This review summarizes and discusses recent studies investigating the anti-inflammatory effects of flavonoids by targeting non-canonical inflammasomes, especially caspase-11 non-canonical inflammasome, and provides new insights into the development of flavonoids and flavonoid-based remedies as potential nutraceuticals that prevent and treat inflammation-related human diseases.

## 2. Flavonoids

### 2.1. General Overview of Flavonoids

Flavonoids are an important class of phytochemicals with polyphenolic structures and are ubiquitously present as secondary metabolites in plants, fruits, vegetables, and beverages. In recent years, interest in flavonoids as bioactive compounds that play pharmacological roles in various human diseases, which can be developed as pharmaceutical leads, has exponentially increased. Studies have reported that flavonoids play an essential role in protecting cells from oxidative stress [60,61]. Several studies have reported that flavonoid-mediated antioxidative activity results in anticancer effects in various types of cancers, including gastric, liver, breast, prostate, cervical, pancreatic, brain, and blood cancers [62,63,64,65,66,67,68,69,70,71]. Flavonoids have been demonstrated to play a protective role in metabolic diseases, such as diabetes mellitus, obesity [72,73,74,75], and cardiovascular diseases [41,76,77,78]. Flavonoids also exert multiple neuroprotective activities in the brain by protecting neurons against neurotoxins; inhibiting neuroinflammation and neurodegeneration; and increasing memory, cognitive, and learning function [79,80,81]. Moreover, many studies have successfully demonstrated that flavonoids play an anti-inflammatory role in inflammatory responses and diseases [47,48,49]. Early studies demonstrating the anti-inflammatory roles of flavonoids focused on the priming process, which is an inflammation-preparing step [50,51,52,53,54]. Interestingly, recent studies have further shown that flavonoids also play an anti-inflammatory role by targeting inflammasome activation during the triggering process, which is an inflammation-activating step in inflammatory responses and diseases [49,55,56,57,58,59], strongly suggesting that flavonoids are natural pharmacological compounds with anti-inflammatory activity by targeting both priming and triggering processes in inflammatory responses and diseases.

### 2.2. Structure and Classification of Flavonoids

More than 10,000 compounds belong to the flavonoid family [82]. Flavonoids have the common structure of a 15-carbon C6–C3–C6 skeleton consisting of two phenyl rings, known as the A and B rings, and one heterocyclic ring, known as the C ring, containing oxygen (Figure 1A). Flavonoids can be classified into different subgroups, such as flavones, flavonols, flavanones, flavanols, isoflavones, leucoanthocyanidins, anthocyanidins, and chalcones, depending on the position of the linkage between rings B and C, oxidation of the C ring, and degree of unsaturation (Figure 1B) [83,84]. The rings can be modified by hydrogenation, hydroxylation, methylation, malonylation, sulfation, and glycosylation, which can exert different biological and pharmacological effects [83,84].

Flavones consist of a backbone of 2-phenylchromen-4-one bearing a phenyl substituent at position 2 (Figure 1B). Flavones are widely found in leaves, flowers, and fruits, and luteolin, apigenin, tangeritin, chrysin, and 6-hydroxyflavone are flavonins.

Isoflavones are isomers of flavones that differ from flavones in the location of the phenyl group. Flavones are chromones substituted with a phenyl group at the 2-position, whereas isoflavones have a phenyl group at the 4-position of the C ring (Figure 1B). The most common sources of isoflavones are soybeans and leguminous plants, and the major isoflavones in soybeans are genistein and daidzein. Isoflavones are phytoestrogens that exert pharmacological effects on various hormonal and metabolic diseases [85].

Flavonols have a 3-hydroxyflavone backbone with a hydroxyl group at position 3 of the C ring and are diverse at different positions in the patterns of glycosylation, methylation, and hydroxylation (Figure 1B). Various vegetables, fruits, teas, and red wine are rich sources of flavonols. Quercetin, kaempferol, morin, myricetin, and fisetin belong to this subclass of flavonoids.

Flavanones, also known as dihydroflavones, have the same structure, but the C ring is saturated between positions 2 and 3 (Figure 1B). Flavanones are generally present in many citrus fruits and are responsible for their bitter taste. Many flavanones, such as hesperidin, hesperetin, narirutin, naringenin, naringin, and eriodictyol, have been discovered over the past decade. Interestingly, flavanones have been demonstrated to have various pharmacological activities, including antioxidative, anti-inflammatory, and antiallergic effects [86,87].

Flavanols, also known as flavan-3-ols, are derivatives of flavans that possess a 2-phenyl-3,4-dihydro-2H-chromen-3-ol backbone and have a saturated C ring between 2 and 3 (Figure 1B). Flavanols are abundant in some fruits and include a wide range of compounds, such as catechin, epicatechin gallate, epigallocatechin, epigallocatechin gallate, proanthocyanidins, theaflavins, and thearubigins.

Flavanonols, also known as dihydroflavonols or catechins, consist of the backbone of 3-hydroxy-2,3-dihydro-2-phenylchromen-4-one and are 3-hydroxy derivatives of flavanones (Figure 1B). Flavanonols are highly diversified and multi-substituted in structure, and like flavanones and flavanols, they have a saturated C ring between 2 and 3 (Figure 1B). Flavanonols are found in some plants, such as *Myrsine seguinii*, *Paepalanthus argenteus*, and *Smilax glabra*, [88,89] and include xeractinol, taxifolin, aromadendrin, and engeletin.

Anthocyanins are flavonoids with the most complicated chemical structure and are based on the chemical structure of the flavylium cation with various substituted groups of hydrogen atoms (Figure 1B). Anthocyanins are predominantly found in various fruits and flowers and are responsible for their color. More than 30 anthocyanins have been identified, including cyanidin, delphinidin, malvidin, pelargonidin, peonidin, and petunidin.

Leucoanthocyanidins are a group of derivatives of anthocyanidins and anthocyanins that possess the structure of flavan-3,4-diols (Figure 1B). Leucoanthocyanidins have been identified as intermediates in anthocyanidin biosynthesis in flowers [90] and are found in *Anadenanthera peregrina* and several species of *Nepenthes* and *Acacia*. Leucoanthocyanidins include leucocyanidin, leucodelphinidin, leucofisetinidin, leucomalvidin, leucopelargonidin, leucopeonidin, leucorobinetinidin, melacacidin, and teracacidin.

Chalcones have a unique structure characterized by the absence of the C ring of the basic flavonoid skeleton, and are referred to as open-chain flavonoids (Figure 1B). Chalcones are found in some fruits and vegetables as well as in certain wheats. The major chalcones include phloridzin, arbutin, phloretin, and chalconaringenin.

## 3. Caspase-11 Non-Canonical Inflammasome

### 3.1. Discovery and Structure

Numerous studies have investigated the roles of canonical inflammasomes in innate immune responses stimulated by various PAMPs and DAMPs. Inflammatory responses are highly activated in response to cholera toxin B in an NLRP3 inflammasome-dependent manner in LPS-primed macrophages, and this inflammatory response is abolished in macrophages derived from the mouse strain 129S6, which expresses a truncated nonfunctional caspase-11 protein [9]. This strain of 129S6 mice is also much more resistant to the lethal dose of LPS that induces acute septic shock [9], which suggests that caspase-11 is different from the canonical inflammasomes and plays a unique role in inflammatory responses in macrophages with a molecular mechanism distinct from that of the canonical inflammasomes. Follow-up studies have successfully established that caspase-11-mediated inflammatory responses are activated by the caspase-11 inflammasome, which does not belong to canonical inflammasomes; therefore, this inflammasome was named the caspase-11 non-canonical inflammasome [9].

Caspase-11 belongs to a group of inflammatory caspases that is distinguished from a group of apoptotic caspases. Caspase-11 is an intracellular PRR consisting of an amino-terminal caspase recruitment domain (CARD), followed by two catalytic domains: a p20 large catalytic domain and a carboxyl-terminal p10 small catalytic domain (Figure 2A). Caspase-11 was initially discovered in mice, and many studies have attempted to identify human caspase-11. Unexpectedly, human caspase-11 has not yet been identified; however, numerous studies have confirmed that caspase-4/5 are homologs of mouse caspase-11 [12,13,15,91]. Human caspase-4/5 has a domain structure similar to that of mouse caspase-11, but their amino acid lengths are different; mouse caspase-11 and human caspase-4/5 are 373, 377, and 434 amino acids in length, respectively (Figure 2A).

### 3.2. Caspase-11 Non-Canonical Inflammasome-Activated Inflammatory Signaling Pathways

Canonical inflammasomes are activated in response to a variety of PAMPs and DAMPs [1,2]. However, unlike canonical inflammasomes, LPS is the only PAMP that activates non-canonical inflammasomes [13,14,15,16]. LPS is an endotoxin found in the cell walls of gram-negative bacteria. Extracellular LPS derived from gram-negative bacteria enters host cells via endocytosis mediated by cell surface receptors, such as Toll-like receptor 4 and receptor for advanced glycation end-product [17]. Extracellular LPS also enters the host cells via bacterial outer membrane vesicle-mediated internalization [17]. LPS is released from internalized endosomes or vacuoles containing intracellular gram-negative bacteria. Guanylate-binding proteins (GBPs) are interferon (IFN)-inducible GTPase family members that are expressed in response to IFN stimulation. The GBPs bind with endosomes and vacuoles and consequently disrupt their membrane integrity, leading to the release and cytosolic access of LPS to mouse caspase-11 and human caspase-4/5 [92,93,94,95].

Caspase-11 senses intracellular LPS via direct interactions. This direct interaction is mediated by the binding of the CARDs of the caspase-11 with the lipid A moiety of LPS, which is a highly conserved component of LPS, resulting in the formation of LPS-caspase-11 complexes (Figure 2B) [13,14,15,16]. The caspase-11 non-canonical inflammasome is then formed by oligomerization of LPS-caspase-11 complexes through CARD-CARD interaction, and the caspase-11 non-canonical inflammasome is subsequently activated by auto-proteolysis (Figure 2B) [96]. Auto-proteolysis is a key determinant of caspase-11 non-canonical inflammasome activation. Activation of the caspase-11 non-canonical inflammasome is mediated by auto-proteolysis at the 285 aspartic acid residue (Asp_285_) of caspase-11, and the 254 cysteine residue (Cys_254_) of caspase-11 has been identified as a critical residue that has enzymatic activity that triggers Asp_285_ auto-proteolysis (Figure 2B) [97].

Activation of the caspase-11 non-canonical inflammasome induces two main inflammatory signaling pathways by activating several downstream effector molecules [13,14,15,16]. Caspase-11 non-canonical inflammasome activation directly promotes proteolytic processing of GSDMD at the 276 asparagine residue (Asp_276_) to produce both N-GSDMD and carboxyl-terminal GSDMD fragments. N-GSDMD then moves to the cell membranes and generates GSDMD pores in them, leading to cell swelling and osmotic rupture, known as pyroptosis. Caspase-11 non-canonical inflammasome activation also promotes proteolytic activation of caspase-1, and the active form of caspase-1 subsequently induces proteolytic maturation and secretion of the pro-inflammatory cytokines IL-1β and -18 through GSDMD pores. Interestingly, the caspase-11 non-canonical inflammasome indirectly activates caspase-1 through functional interplay with the NLRP3 canonical inflammasome. The direct interaction between caspase-11 non-canonical and NLRP3 canonical inflammasomes potentiates the activation of the NLRP3 canonical inflammasome, leading to the proteolytic activation of caspase-1 [98]. Caspase-11 non-canonical inflammasome also indirectly activates the NLRP3 canonical inflammasome. Potassium ion (K^+^) efflux is a key event in the activation of the NLRP3 canonical inflammasome, and caspase-11 non-canonical inflammasome activation induces potassium ion (K^+^) efflux through pyroptosis-mediated cell membrane damage and membrane gate proteins, such as P_2_X7 channels, bacterial pore-forming toxins, and pannexin 1 channels [17,18,19,20]. These results strongly suggest that canonical and non-canonical inflammasomes play a cooperative rather than an independent role in inflammasome-activated inflammatory signaling pathways. The caspase-11 non-canonical inflammasome-activated inflammatory signaling pathway is shown in Figure 2C.

## 4. Flavonoid-Mediated Anti-Inflammatory Roles by Targeting Caspase-11 Non-Canonical Inflammasome

Many studies have demonstrated the anti-inflammatory roles of various flavonoids in inflammatory responses and diseases by suppressing the activation of canonical inflammasomes, particularly the NLRP3 canonical inflammasome [49,55,56,57,58,59]. Interestingly, a growing number of studies have also reported that flavonoids exert strong anti-inflammatory activity by inhibiting the activation of the caspase-11 non-canonical inflammasome, which is a key player in inflammatory responses and various immunopathological conditions. Here, we summarize and discuss recent studies that have investigated the anti-inflammatory role of various flavonoids in inflammatory responses and diseases.

### 4.1. Luteolin

Luteolin is a 3′,4′,5,7-tetrahydroxyflavone (Figure 3A) found in various vegetables, fruits, flowers, and medicinal plants, and plays an anti-inflammatory role by decreasing the production of inflammatory mediators and pro-inflammatory cytokines [99]. Luteolin also exerts an anti-inflammatory effect by inhibiting the activation of the NLRP3 canonical inflammasome in macrophages [100,101]. Recently, Hwang et al. demonstrated the in vitro and in vivo anti-inflammatory roles of luteolin by targeting the caspase-11 non-canonical inflammasome in macrophages. Luteolin in *Viburnum pichinchense* inhibited caspase-11 non-canonical inflammasome-activated pyroptosis and IL-1β production in macrophages [102]. An in vivo study further showed that luteolin-containing *V. pichinchense* ameliorated HCl/EtOH-induced gastritis in mice [102], suggesting that the ameliorative effect of luteolin on gastritis may be mediated by inhibiting the activation of the caspase-11 non-canonical inflammasome in macrophages. Yan et al. investigated luteolin-inhibited caspase-4/11 non-canonical inflammasome activation in sepsis. Luteolin inhibited in vitro activity of human caspase-4, pyroptosis, and the secretion of the pro-inflammatory cytokines IL-1β, -16, and -1α in macrophages [103]. Luteolin also suppresses LPS-induced lethal sepsis in mice [103]. These results indicated that luteolin suppresses inflammatory responses by targeting human caspase-4 and mouse caspase-11 non-canonical inflammasomes in macrophages, which can protect against endotoxin-stimulated lethal sepsis. Zhang et al. demonstrated an inhibitory effect of luteolin on caspase-11 non-canonical inflammasome activation in sepsis-induced lung injury. Luteolin reduces the serum levels of pro-inflammatory cytokines and alleviates caspase-11 non-canonical inflammasome-activated pyroptosis in the lung tissues of cecal ligation and puncture (CLP)-induced acute lung injury (ALI) in mice [104]. Luteolin also attenuates CLP-induced ALI in mice [104], which strongly suggests that it exerts an inhibitory action on pro-inflammatory cytokine production and lung pyroptosis by inhibiting the caspase-11 non-canonical inflammasome and, as a result, ameliorates sepsis-induced ALI. Taken together, luteolin has strong anti-inflammatory activity by targeting human caspase-4 and mouse caspase-11 non-canonical inflammasomes in inflammatory responses and immunopathologies, such as gastritis, sepsis, and sepsis-induced ALI.

### 4.2. Scutellarin

Scutellarin is a 4′,5,6-hydroxyflavone-7-glucuronide (Figure 3B) that is frequently found in the genera *Scutellaria* (Lamiaceae) and *Erigeron* (Asteraceae) and has long been used in traditional Chinese medicine. Scutellarin has been demonstrated to show various pharmacological activities for neurodegenerative, metabolic, infectious, and cardiovascular diseases as well as cancers [105,106]. Studies have also demonstrated the anti-inflammatory activity of scutellarin by inhibiting canonical inflammasomes, particularly the NLRP3 canonical inflammasome, in inflammatory responses and various diseases [107,108,109,110,111,112]. Recently, Ye et al. reported an inhibitory role of scutellarin in non-canonical inflammasome-activated inflammatory responses in macrophages. Scutellarin suppressed LPS-stimulated proteolytic activation of caspase-11 and GSDMD, resulting in reduced pyroptosis and IL-1β secretion in bone marrow-derived macrophages and J774A.1 macrophages [113]. Scutellarin also inhibits NLRP3 canonical inflammasome activation, but scutellarin-mediated inhibition of caspase-11 non-canonical inflammasome activation is independent of NLRP3 canonical inflammasome pathways in macrophages [113], suggesting that scutellarin simultaneously inhibits both caspase-11 non-canonical and NLRP3 canonical inflammasomes, leading to reduced pyroptosis and IL-1β secretion in macrophages. Peng et al. also reported the role of scutellarin in inflammasome-activated inflammatory responses and idiopathic pulmonary fibrosis (IPF). Inflammatory responses, including the elevated expression of NLRP3, caspase-11, caspase-1, ASC, GSDMD, and pro-inflammatory cytokines IL-1β and -18, were significantly induced in the lung tissues of bleomycin-induced pulmonary fibrosis mice [108]. Interestingly, scutellarin alleviated lung damage and suppressed inflammatory responses, except for the increased expression of caspase-11 [108], indicating that caspase-11 non-canonical inflammasome activation may not be the key molecule in scutellarin-mediated inhibitory effects on inflammatory responses and IPF pathogenesis. Given the evidence from these studies, scutellarin is an anti-inflammatory flavonoid that may selectively inhibit caspase-11 non-canonical inflammasome-activated inflammatory responses depending on the disease type.

### 4.3. Apigenin

Apigenin is a 4′,5,7-trihydroxyflavone (Figure 3C) found in a wide variety of fruits, vegetables, chamomile teas, and medicinal herbs. Apigenin presents multiple pharmacological activities, including antioxidative, anticardiovascular, antidiabetic, neuroprotective, and anticancer activities [114,115,116,117,118]. Apigenin has also been demonstrated to exert anti-inflammatory effects by inhibiting inflammatory mediators, pro-inflammatory cytokines, cell adhesion molecules, and signaling molecules [119,120,121] and by ameliorating various immunopathological conditions [120,121,122]. Numerous studies have further demonstrated apigenin-mediated anti-inflammatory action in the activation of canonical inflammasomes, particularly the NLRP3 inflammasome [56,123,124,125]; however, studies demonstrating the non-canonical inflammasome-inhibited anti-inflammatory role of apigenin have been very limited. A recent study reported the anti-inflammatory role of apigenin in inflammatory bowel disease by targeting the caspase-11 non-canonical inflammasome. Dietary apigenin ameliorated colon damage in dextran sulfate sodium-induced colitis in mice [124]. Apigenin also inhibits the proteolytic activation of caspase-11 and -1, resulting in decreased secretion of IL-1β and -18 in the colon tissues of mice with colitis [124]. These results strongly suggest that apigenin is an anti-inflammatory flavonoid that protects against the development of chronic ulcerative colitis by targeting caspase-11 non-canonical inflammasome activation, and as a result, inhibits the production of downstream pro-inflammatory cytokines.

### 4.4. Epigallocatechin-3-Gallate (EGCG)

Epigallocatechin-3-gallate (EGCG), a type of catechin, is a gallate ester obtained by the formal condensation of gallic acid with the (3R)-hydroxy group of (−)-epigallocatechin (Figure 3D). EGCG is most abundant in teas and is also found in fruits, such as apples and plums; vegetables, such as onions; and nuts, such as pecans and hazelnuts. EGCG has therapeutic potential against various pathological conditions, such as neurodegenerative, cardiovascular, and infectious diseases; obesity; diabetes; oxidative stress; and cancer [126]. EGCG has anti-inflammatory activity and therapeutic potential against chronic inflammatory diseases, such as rheumatoid arthritis, gouty arthritis, and systemic lupus erythematosus, by targeting various inflammatory molecules and pro-inflammatory cytokines [58,126]. EGCG also plays an anti-inflammatory role and ameliorates some chronic inflammatory diseases by inhibiting canonical inflammasomes, such as NLRP1, NLRP3, and AIM2 [56,127,128,129,130]. An interesting study further reported the anti-inflammatory role of EGCG through the inhibition of the non-canonical inflammasome in microglial inflammation and neurotoxicity. EGCG decreases LPS/A-stimulated inflammation and neurotoxicity in microglial cells [129]. EGCG reduced the proteolytic activation of caspase-11 and further inhibited caspase-11 non-canonical inflammasome-activated secretion of IL-1β and IL-18 in LPS/A-stimulated microglial cells [129]. These results suggest that EGCG attenuates microglial inflammation-mediated neurotoxicity by inhibiting the activation of the caspase-11 non-canonical inflammasome and subsequent production of pro-inflammatory cytokines in microglial cells.

### 4.5. Quercetin

Quercetin is a 3,3′,4′,5,7-pentahydroxyflavone (Figure 3E) that occurs naturally in various fruits and vegetables. Quercetin is a strong antioxidant that belongs to the flavonol group and is generally present in the glycoside form. Quercetin and its derivatives show promising pharmacological effects, including antioxidant, antidiabetic, antimicrobial, neuroprotective, anticardiovascular, and anticancer effects [131,132,133,134,135,136,137,138]. Quercetin is one of the most studied flavonoids in inflammatory responses and diseases [139,140,141,142]. Quercetin also plays an anti-inflammatory role by inhibiting inflammasomes, particularly the NLRP3 canonical inflammasome [143,144,145,146]. Recently, an interesting study demonstrated that quercetin plays an anti-inflammatory role by inhibiting the caspase-11 non-canonical inflammasome in macrophages and in gastritis. Quercetin in *V. pichinchense* ameliorated HCl/EtOH-induced gastritis in mice and inhibited caspase-11 non-canonical inflammasome-activated pyroptosis and IL-1β secretion in RAW264.7 macrophages [102]. These results indicated that quercetin in *V. pichinchense* plays an anti-inflammatory role and ameliorates gastritis by inhibiting the caspase-11 non-canonical inflammasome in macrophages.

### 4.6. Kaempferol

Kaempferol is a 3,4′,5,7-tetrahydroxyflavone (Figure 3F) and a natural flavonol abundantly found in a variety of plants, such as *Pteridophyta*, *Pinophyta*, and *Angiospermae*; plant-originating foods, such as broccoli, spinach, kale, beans, and tea; and fruits, such as apples, grapes, tomatoes, and peaches [147]. Kaempferol has been demonstrated to have pharmacological activity in various immunopathological conditions, including infectious, neuronal, cardiovascular, and metabolic diseases, as well as cancers [147,148,149,150,151,152]. Similar to quercetin, kaempferol is a flavonoid that exhibits a strong anti-inflammatory effect and has potential as an anti-inflammatory therapeutic [153,154,155,156,157]. Kaempferol also suppresses canonical inflammasomes, and as a result, alleviates numerous inflammatory diseases [56,158,159,160]. However, few studies have reported the anti-inflammatory role of kaempferol by inhibiting non-canonical inflammasomes. A recent study reported kaempferol-mediated anti-inflammatory activity by inhibiting the caspase-11 non-canonical inflammasome in macrophages. Kaempferol in *V. pichinchense* suppressed caspase-11 inflammasome activation, leading to reduced pyroptosis and IL-1β secretion in RAW264.7 macrophages [102]. Kaempferol also alleviated HCl/EtOH-induced gastritis in mice [102], suggesting that, similar to quercetin, kaempferol mitigates gastritis by targeting caspase-11 non-canonical inflammasome in macrophages, which provides evidence that quercetin and kaempferol might be promising anti-inflammatory therapeutics against gastritis by targeting both canonical and non-canonical inflammasomes in macrophages.

### 4.7. Icariin

Icariin is a 7-(β-D-glucopyranosyloxy)-5-hydroxy-4′-methoxy-8-(3-methylbut-2-en-1-yl)-3-(α-L-rhamnopyranosyloxy) flavone, which is an 8-prenyl derivative of kaempferol (Figure 3G). Icariin is a natural flavonoid found in several plant species belonging to the *Epimedium* genus. Icariin has biological roles and pharmacological activities, including antiaging, antiosteoporosis, antioxidative, antiatherosclerotic, and anticancer activities [161,162,163,164,165]. Additionally, icariin exerts anti-inflammatory effects by inhibiting the priming step and canonical inflammasomes in inflammatory responses and diseases [166,167,168,169,170,171]. Icariin also plays an anti-inflammatory role by inhibiting non-canonical inflammasomes in LPS-stimulated inflammatory responses [172]. Icariin and phosphorylated icariin reduced LPS-induced inflammatory responses and decreased the expression of caspase-4, a human homolog of mouse caspase-11, in LPS-stimulated human LS174T intestinal goblet cells [172]. These results indicated that icariin and phosphorylated icariin alleviate LPS-induced inflammatory responses by targeting the caspase-4 non-canonical inflammasome in human intestinal goblet cells.

### 4.8. Baicalin

Baicalin is a 7-D-glucuronic acid-5,6-dihydroxyflavone that belongs to the flavone subgroup (Figure 3H) and is found in several species of the genus *Scutellaria*, including *Scutellaria baicalensis* and *Scutellaria lateriflora*. Baicalin is the major metabolite of baicalein originally isolated from *S. baicalensis*. Baicalin has significant antiviral, antibacterial, antioxidative, and anticancer activities [173,174,175,176]. Baicalin attenuates inflammatory responses and ameliorates inflammatory diseases by modulating various inflammatory signaling pathways, including the NLRP3 canonical inflammasome-activated signaling pathways [177,178,179,180,181,182,183]. Baicalin also plays an anti-inflammatory role by targeting non-canonical inflammasomes and has been demonstrated to protect against mycotoxin-induced liver and kidney injury by inhibiting the caspase-11 non-canonical inflammasome. Baicalin ameliorated zearalenone (ZEA)-induced inflammation and pathological changes in the liver and kidneys of chicks [184]. Baicalin decreased the ZEA-induced expression of caspase-11 and inflammatory cytokines in the liver of chicks [184], suggesting that baicalin attenuates mycotoxin-induced inflammation and tissue injury by inhibiting the caspase-11 non-canonical inflammasome and the subsequent production of pro-inflammatory cytokines.

### 4.9. Morin

Morin, a 2′,3,4′,5,7-pentahydroxyflavone (Figure 3I), is a natural pigment obtained from the *Moraceae* family. Morin is associated with numerous pharmacological properties, such as antimicrobial, antioxidative, antidiabetic, anticancer, and tissue-protective effects, and has been widely used in the treatment of various human diseases [185]. Morin has also been reported to have anti-inflammatory activity, with neuroprotective, hepatoprotective, gastroprotective, and articular protective effects in various inflammatory diseases [186,187,188,189,190]. Similar to a study demonstrating the baicalin-mediated inhibitory effect on liver and kidney injury by targeting the caspase-11 non-canonical inflammasome [184], a recent study also reported that morin plays a protective role in toxin-induced liver and kidney injury by inhibiting the caspase-11 non-canonical inflammasome. Morin alleviated aflatoxin B1 (AFB1)-induced liver and kidney damage in chicks [191]. Further mechanistic studies revealed that morin suppressed the production of caspase-11, pro-inflammatory cytokines, and inflammatory factors, resulting in the inhibition of caspase-11 non-canonical inflammasome-induced inflammatory responses in AFB1-stimulated livers [191]. These results suggest that morin reduces toxin-induced inflammatory responses and protects against inflammatory liver and kidney injury by inhibiting the caspase-11 non-canonical inflammasome and downstream pro-inflammatory cytokines.

### 4.10. Naringenin

Naringenin is a 4′,5,7-trihydroxyflavonone (Figure 3J) and within the flavonoid groups, it is a flavanone derived from naringin or narirutin. Naringenin is predominantly present in a variety of citrus fruits, such as grapefruits, oranges, herbs, and tomatoes. Naringenin has many pharmacological properties, including antimicrobial, antiaging, antiasthma, antidiabetic, antihyperlipidemic, antioxidative, anticancer, neuroprotective, cardioprotective, and hepatoprotective effects [192]. Moreover, naringenin exhibits anti-inflammatory properties by targeting various signaling pathways involved in priming-induced and canonical inflammasome-activated inflammatory responses, leading to the attenuation of a wide range of immunopathological conditions [192,193,194,195]. Recently, an interesting study reported the anti-inflammatory and protective roles of naringenin in ER stress-induced renal ischemia/reperfusion (I/R) injury by targeting non-canonical inflammasomes. Naringenin ameliorated renal I/R injury by improving renal function and attenuating renal tissue damage in mice [196]. Naringenin also significantly reduced the generation of caspase-4 and -11 as well as proteolytic cleaved GSDMD, resulting in the inhibition of pyroptosis and apoptosis in the renal tissues of I/R mice and hypoxia/reoxygenation (H/R)-exposed HK-2 cells [196]. These results suggest that naringenin has strong anti-inflammatory properties and protects renal tissues against I/R injury by inhibiting the caspase-11 non-canonical inflammasome and inflammasome-activated pyroptosis.

## 5. Conclusions

Flavonoids are naturally occurring bioactive compounds that modulate many biological activities. Considerable efforts have been made to elucidate the protective and pharmacological roles of flavonoids in a wide range of human immunopathologies. However, many previous studies have demonstrated that these effects are mediated by flavonoids, mainly focusing on the priming step of inflammatory responses. In addition, despite numerous studies focusing on the triggering step of inflammatory responses, the effects of flavonoids have focused heavily on canonical inflammasomes, particularly the NLRP3 inflammasome [55,56,58], which has prompted questions regarding the pharmacological roles of flavonoids in inflammatory responses and diseases induced by the activation of non-canonical inflammasomes, such as mouse caspase-11 and human caspase-4 non-canonical inflammasomes. Interestingly, recent studies have provided substantial evidence to support the new anti-inflammatory roles of flavonoids in inflammatory responses and diseases by targeting non-canonical inflammasomes, as summarized in Table 1. Despite these successful studies, there remain several limitations in understanding the anti-inflammatory roles of flavonoids in non-canonical inflammasome-activated inflammatory responses and diseases. First, as summarized in Table 1, most studies have used mouse cells and animal disease models, particularly mouse disease models, in which flavonoids target the mouse caspase-11 non-canonical inflammasome rather than the human caspase-4 non-canonical inflammasome in inflammatory responses and diseases. This is unavoidable because non-canonical inflammasomes were first discovered in mice, and studies should prove the pharmacological effects of flavonoids on inflammatory diseases using animal models before using patients. However, the pharmacological roles of flavonoids in inflammatory diseases should be investigated in patients by targeting the human caspase-4 non-canonical inflammasome. Second, previous studies have been limited to several inflammatory diseases, such as gastritis, colitis, endotoxemia, and organ injuries. Given the evidence that non-canonical inflammasome-activated inflammatory responses share common molecular mechanisms, future studies should be extended to cover a larger number of inflammatory diseases. Third, although over 6000 naturally occurring flavonoids have been identified [84], only a few have been demonstrated to attenuate non-canonical inflammasome-associated inflammatory responses and diseases. Since substantial evidence has emphasized that non-canonical inflammasomes are key players in inducing inflammation, leading to the exacerbation of multiple inflammatory and infectious diseases [13,14,15,24,37,197,198,199,200], further studies to identify new flavonoids targeting non-canonical inflammasomes and to demonstrate their pharmacological roles in diseases associated with non-canonical inflammasomes need to be undertaken. Fourth, most of the studies demonstrated the preventive effect of flavonoids in the inflammatory responses and diseases, and the therapeutic effect of flavonoids should also be examined in the inflammatory responses and diseases. Finally, as discussed in this review, flavonoids effectively ameliorate the inflammatory responses and various inflammatory diseases mediated by the activation of caspase-11 non-canonical inflammasome, which strongly suggests that flavonoids could be potential nutraceuticals for the treatment of inflammatory diseases exacerbated by the activation of caspase-11 non-canonical inflammasome. Therefore, the importance of the identification and validation of more flavonoids that have the pharmacological activity for the inflammatory diseases exacerbated by the activation of caspase-11 non-canonical inflammasome cannot be emphasized enough. In conclusion, this review discusses the emerging anti-inflammatory roles of flavonoids in inflammatory responses and multiple immunopathologies induced by non-canonical inflammasomes, as summarized in Figure 4. This review improves current knowledge of the new anti-inflammatory roles of flavonoids and provides insights into the development of flavonoids as nutraceuticals to prevent and treat a variety of human diseases associated with non-canonical inflammasomes.

## Figures and Tables

**Figure 1 ijms-24-10402-f001:**
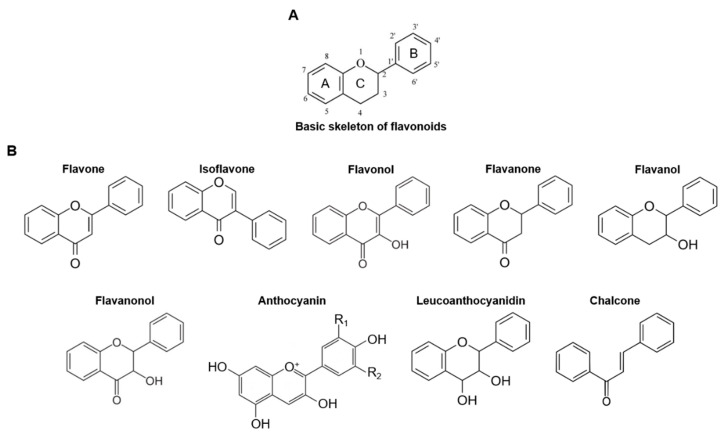
The structure of flavonoids. (**A**) The basic backbone of flavonoids. (**B**) The chemical structure of flavonoid subgroups: flavone, isoflavone, flavonol, flavanone, flavanol, flavanonol, anthocyanin, leucoanthocyanidin, and chalcone.

**Figure 2 ijms-24-10402-f002:**
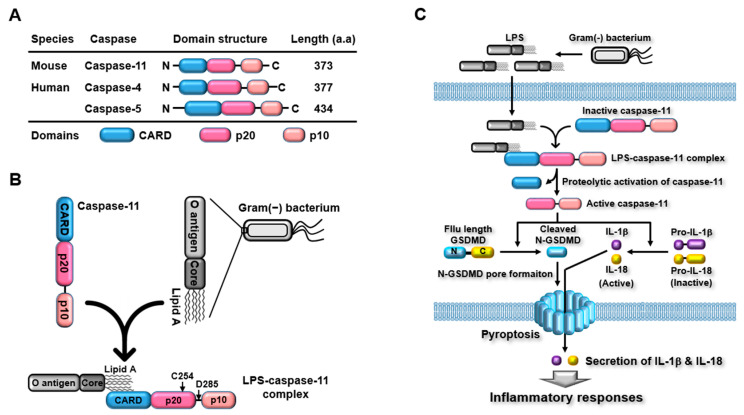
Caspase-11 non-canonical inflammasome-activated inflammatory signaling pathways. (**A**) Comparison of the structure of mouse caspase-11 and human caspase-4 and -5. Mouse caspase-11 and human caspase-4 and -5 have the same domains (N-terminal CARD, p20, and C-terminal p10) with different amino acid lengths. (**B**) Direct recognition of LPS by caspase-11. Caspase-11 recognizes LPS by direct interaction between the CARD of caspase-11 and lipid A of LPS, leading to the formation of an LPS-caspase-11 complex. C254 is the 254 cysteine residue of caspase-11 that has auto-proteolytic enzymatic activity. D285 is the 285 aspartic acid residue of caspase-11 that undergoes auto-proteolysis. (**C**) Caspase-11 non-canonical inflammasome-activated inflammatory signaling pathways.

**Figure 3 ijms-24-10402-f003:**
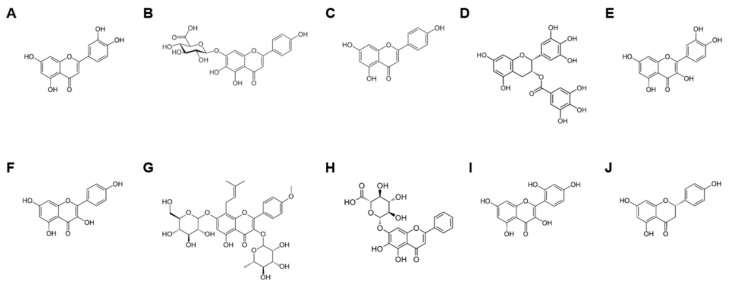
The chemical structure of the flavonoids discussed in this review. (**A**) Luteolin, (**B**) scutellarin, (**C**) apigenin, (**D**) epigallocatechin-3-gallate, (**E**) quercetin, (**F**) kaempferol, (**G**) icariin, (**H**) baicalin, (**I**) morin, and (**J**) naringenin.

**Figure 4 ijms-24-10402-f004:**
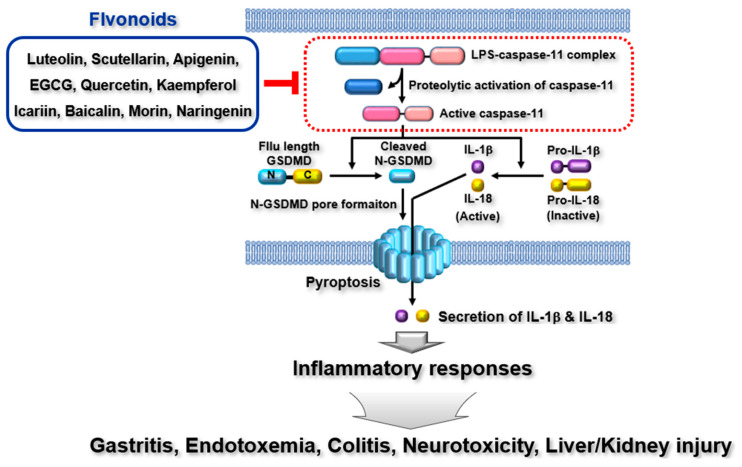
Schematic summary of flavonoid-mediated ameliorative properties in inflammatory responses and immunopathologies by targeting non-canonical inflammasomes.

**Table 1 ijms-24-10402-t001:** Flavonoid-mediated anti-inflammatory roles by targeting the caspase-11 non-canonical inflammasome.

Flavonoids	Diseases	Roles	Models	Ref.
Luteolin	Gastritis	Luteolin in *Viburnum pichinchense* inhibited caspase-11 non-canonical inflammasome-activated pyroptosis and IL-1β production in macrophages *Viburnum pichinchense* containing luteolin ameliorated HCl/EtOH-induced gastritis in mice	RAW264.7 cells HCl/EtOH-induced gastritis mice	[102]
	Sepsis	Luteolin inhibited in vitro activity of human caspase-4 Luteolin reduced pyroptosis and the secretion of IL-1β, IL-16, and IL-1α in macrophages Luteolin suppressed LPS-induced lethal sepsis in mice	RAW264.7, THP-1 cells LPS-induced sepsis mice	[103]
	ALI	Luteolin reduced the serum levels of pro-inflammatory cytokines in the lung tissues of CLP-induced ALI in mice Luteolin alleviated caspase-11 non-canonical inflammasome-activated pyroptosis in the lung tissues of CLP-induced ALI in mice Luteolin attenuated CLP-induced ALI in mice	CLP-induced ALI mice	[104]
Scutellarin	Inflammatory response	Scutellarin suppressed LPS-stimulated proteolytic activation of caspase-11 and GSDMD in macrophages Scutellarin reduced pyroptosis and IL-1β secretion in macrophages Scutellarin inhibited NLRP3 canonical inflammasome activation Scutellarin-mediated inhibition of caspase-11 non-canonical inflammasome activation was independent of NLRP3 canonical inflammasome pathways in macrophages	BMDMs, J774A.1, RAW264.7 cells	[113]
	IPF	Expression of NLRP3, caspase-11, caspase-1, ASC, GSDMD, IL-1β, and IL-18 significantly increased in the lung tissues of bleomycin-induced pulmonary fibrosis mice. Scutellarin alleviated the lung damage of bleomycin-induced pulmonary fibrosis mice. Scutellarin suppressed the inflammatory responses except for the increased expression of caspase-11 in the lung tissues of bleomycin-induced pulmonary fibrosis mice	Bleomycin-induced pulmonary fibrosis mice	[108]
Apigenin	Colitis	Apigenin ameliorated colon damage in DSS-induced colitis mice Apigenin inhibited proteolytic activation of caspase-11 and -1 in the colon tissues of colitis mice Apigenin decreased the secretion of IL-1β and IL-18 in the colon tissues of colitis mice	DSS-induced colitis mice	[124]
EGCG	Inflammatory response	EGCG decreased LPS/Aβ-stimulated inflammation and neurotoxicity in microglial cells EGCG reduced proteolytic activation of caspase-11 in LPS/Aβ-stimulated microglial cells EGCG inhibited the caspase-11 non-canonical inflammasome-activated secretion of IL-1β and IL-18 in LPS/Aβ-stimulated microglial cells	BV-2, SH-SY5Y cells	[129]
Quercetin	Gastritis	Quercetin in *Viburnum pichinchense* ameliorated HCl/EtOH-induced gastritis in mice Quercetin inhibited caspase-11 non-canonical inflammasome-activated pyroptosis and IL-1β secretion in macrophages	RAW264.7 cells HCl/EtOH-induced gastritis mice	[102]
Kaempferol	Gastritis	Kaempferol in *Viburnum pichinchense* suppressed caspase-11 inflammasome-activated pyroptosis and IL-1β secretion in macrophages Kaempferol alleviated HCl/EtOH-induced gastritis in mice	RAW264.7 cells HCl/EtOH-induced gastritis mice	[102]
Icariin	Intestine injury	Icariin and phosphorylated icariin reduced the LPS-induced inflammatory responses in human LS174T intestinal goblet cells Icariin and phosphorylated icariin decreased the expression of caspase-4, a human homolog of mouse caspase-11, in LPS-stimulated human LS174T intestinal goblet cells	LS174T cells	[172]
Baicalin	Liver & kidney injury	Baicalin ameliorated ZEA-induced inflammation and pathologic changes of the liver and kidneys in chicks Baicalin decreased ZEA-induced expression of caspase-11 and inflammatory cytokines in the liver of chicks	ZEA-induced liver and kidney injury chicks	[184]
Morin	Liver & kidney injury	Morin alleviated AFB1-induced liver and kidney damage in chicks Morin suppressed the production of caspase-11, pro-inflammatory cytokines, and inflammatory factors in AFB1-stimulated livers	AFB1-induced liver and kidney injury chicks	[191]
Naringenin	Renal I/R injury	Naringenin refined renal functions and attenuated renal tissue damage in I/R mice Naringenin ameliorated renal I/R injury in mice Naringenin inhibited pyroptosis and apoptosis in renal tissues of I/R mice and H/R-exposed HK-2 cells Naringenin reduced the generation of caspase-4, caspase-11, and proteolytic cleaved GSDMD in renal tissues of I/R mice and H/R-exposed HK-2 cells	HK-2 cells I/R injury mice	[196]

## Data Availability

Not applicable.

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
