# Peer review of "Regulatory Roles of Flavonoids in Caspase-11 Non-Canonical Inflammasome-Mediated Inflammatory Responses and Diseases"

_ijms, 2023, doi:10.3390/ijms241210402_

Round 1

Reviewer 1 Report

This review focuses on the role of flavonoids in the non-classical inflammasome signaling pathway of caspase-11 and their anti-inflammatory activities. At the same time, the regulatory effects of this flavonoid on classical inflammasome were also compared. At present, there are more review papers on the regulators of classical inflammasome, and fewer reviews on non-classical inflammasome, so the perspective selection of this paper is unique. However, the discussion part of the paper is too simple, and few views and insights on the application prospects of flavonoids in non-classical inflammasome signaling pathways need to be further expanded.

Minor editing of English language required

Author Response

Respnse:

Thank you for your comments. As per your comments, applications and prospects of flavonoids in non-classical inflammasome signaling pathways have been added in the Discussion section.

Reviewer 2 Report

In this review, Young-Su Yi highlighted the current status of flavonoids in inflammatory responses mediated by the non-canonical Caspase-11 inflammasome and described several studies demonstrating that flavonoids play an anti-inflammatory role through inhibition of the inflammatory non-canonical inflammasome pathway. I consider that this review is correctly written and can be published. Regarding the figures, I think they can be clearer to facilitate their understanding.

Author Response

Response:

Thank you for your comments. As per your comments, the figures have been modified to make them clearer and to facilitate the readers’ understanding.

Round 2

Reviewer 1 Report

Accept in present form